# Influence of 2 Weeks of Mango Ingestion on Inflammation Resolution after Vigorous Exercise

**DOI:** 10.3390/nu16010036

**Published:** 2023-12-21

**Authors:** Camila A. Sakaguchi, David C. Nieman, Ashraf M. Omar, Renee C. Strauch, James C. Williams, Mary Ann Lila, Qibin Zhang

**Affiliations:** 1Human Performance Laboratory, Department of Biology, Appalachian State University, North Carolina Research Campus, Kannapolis, NC 28081, USA; olsonca@appstate.edu (C.A.S.); williamsjc12@appstate.edu (J.C.W.); 2UNCG Center for Translational Biomedical Research, University of North Carolina at Greensboro, North Carolina Research Campus, Kannapolis, NC 28081, USA; amalsagheer@uncg.edu (A.M.O.); q_zhang2@uncg.edu (Q.Z.); 3Food Bioprocessing and Nutrition Sciences Department, Plants for Human Health Institute, North Carolina State University, North Carolina Research Campus, Kannapolis, NC 28081, USA; rcstrauc@ncsu.edu (R.C.S.); mlila@ncsu.edu (M.A.L.)

**Keywords:** mangoes, exercise, inflammation, metabolites, gallotannins, oxylipins, tandem mass spectrometry

## Abstract

Mangoes have a unique nutrient profile (carotenoids, polyphenols, sugars, and vitamins) that we hypothesized would mitigate post-exercise inflammation. This study examined the effects of mango ingestion on moderating exercise-induced inflammation in a randomized crossover trial with 22 cyclists. In random order with trials separated by a 2-week washout period, the cyclists ingested 330 g mango/day with 0.5 L water or 0.5 L of water alone for 2 weeks, followed by a 2.25 h cycling bout challenge. Blood and urine samples were collected pre- and post-2 weeks of supplementation, with additional blood samples collected immediately post-exercise and 1.5-h, 3-h, and 24 h post-exercise. Urine samples were analyzed for targeted mango-related metabolites. The blood samples were analyzed for 67 oxylipins, which are upstream regulators of inflammation and other physiological processes. After 2 weeks of mango ingestion, three targeted urine mango-related phenolic metabolites were significantly elevated compared to water alone (interaction effects, *p* ≤ 0.003). Significant post-exercise increases were measured for 49 oxylipins, but various subgroup analyses showed no differences in the pattern of change between trials (all interaction effects, *p* > 0.150). The 2.25 h cycling bouts induced significant inflammation, but no countermeasure effect was found after 2 weeks of mango ingestion despite the elevation of mango gut-derived phenolic metabolites.

## 1. Introduction

Intensive exercise training induces physiological stress, including inflammation, oxidative stress, and transient immune dysfunction. Our research group and others have investigated nutrition-based approaches, including increased intake of fruits and polyphenols as countermeasures to these exercise-induced indicators of metabolic stress [1,2,3,4,5,6,7,8]. We have focused on the use of multi-omics methods (i.e., lipidomics, metabolomics, proteomics, genomics, and epigenetics) to capture the complex biochemical interactions that occur in these types of sports nutrition investigations [4,5,7].

Oxylipins are bioactive lipids that are produced via the oxygenation of polyunsaturated fatty acids (PUFAs) and were designated as the primary outcome of this study. Oxylipins are synthesized from cell membrane PUFAs as they are released under tight regulation by phospholipase A2 (PLA2) in response to cell activation from various stress-related stimuli, including exercise [9,10,11,12,13]. Cyclooxygenase (COX), lipoxygenase (LOX), and cytochrome P450 (CYP) enzyme systems metabolize the released PUFAs into oxylipins that act as autocrine and paracrine lipid mediators in numerous physiological processes [9,10,11]. Depending on the metabolic context, oxylipins can function as beneficial signaling agents or mediators of inflammation, immune dysfunction, and disease [12,13,14,15,16,17,18,19,20,21,22,23,24,25,26,27,28,29,30,31]. A significant proportion of the physiological and immune system effects from *n*-6 and *n*-3 PUFAs are mediated through these oxidized metabolites (i.e., lipid mediators) and have emerged as sensitive indicators of metabolic change in nutrition-based interventions [9,11,15,18,28,29,30] or in various disease states [10,20,21,22,23,24,25,26,27]. Exercise-induced muscle tissue injury, inflammation, and metabolic stress prompt an innate immune response. Lipid mediators are involved in initiating, mediating, and resolving this process [16,17,18,29,30,31].

Our research group showed that exercise-induced increases in pro-inflammatory oxylipins can be countered through carbohydrate ingestion (both bananas and sports drinks), with the largest effects seen for CYP-derived lipid mediators [5,29]. We also showed that 2 weeks of ingestion of one cup/d blueberries increased the plasma levels of 24 gut-derived phenolics and strongly countered post-exercise increases in plasma levels of 10 pro-inflammatory oxylipins derived from arachidonic acid (ARA) and CYP (ARA-CYP) [29]. In another study with untrained adults, 18 days intake of one cup/day blueberries was linked to a reduction in pro-inflammatory dihydroxy-9Z-octadecenoic acids (diHOMES) and sustained elevations in docosahexaenoic acid (DHA)- and eicosapentaenoic acid (EPA)-derived anti-inflammatory oxylipins in response to a 90 min bout of unaccustomed exercise [30]. These data indicate that oxylipins are sensitive to both exercise and nutritional influences and that inflammation resolution from vigorous or muscle-damaging exercise can be improved by the intake of carbohydrates and polyphenols from fruits.

Mangoes are high in carotenoids (26 mg/100 g), polyphenols (145 mg/100 g), carbohydrates (15 g/100 g), vitamin C (36 mg/100 g), vitamin A (54 µg RAE/100 g), and folate (43 µg/100 g) [32,33]. The total antioxidant content of mangoes is high, and this popular fruit has been extensively investigated for its health-related effects [34,35,36,37,38,39,40]. There are many types of polyphenols in mangoes, with a predominance of 3,4,5-trihydroxybenzoic acids (gallic acid) and other galloyl derivatives [40]. Like other plant foods, mango polyphenols are poorly absorbed in the human small intestine. After biotransformation by colon bacteria, changes in circulating and urine levels of at least 94 polyphenol-related metabolites have been measured [40]. These gut-derived metabolites from mango ingestion include sulfated, methylated, and glucuronide conjugates of 1,2,3-trihydroxybenzene (pyrogallol) that may exert anti-inflammatory effects as indicated in a limited number of cell culture, animal, and human studies [34,35,36,38]. For example, in one small, uncontrolled study of patients with inflammatory bowel disease, an 8-week intake of 200–400 g/day of mango pulp was linked to decreased plasma levels of cytokines related to neutrophil-induced inflammation (35). Mango polyphenols and gut-derived phenolics may influence the gut microbiome and interact with both gut and liver enzyme systems, including CYP and LOX, to inhibit inflammatory oxylipins [34,35,41,42,43,44].

Although untested within an exercise context, these data support our hypothesis that the intake of mangoes (330 g/day or two cups/day for two weeks) has the potential to counter post-exercise increases in pro-inflammatory oxylipins in trained cyclists. This study investigated the effects of 2 weeks of mango ingestion on moderating exercise-induced inflammation in a randomized crossover trial with 22 male and female cyclists. The influence of 2–4 weeks’ intake of fruit carbohydrates and polyphenols on inflammation resolution following exercise stress is an emerging area of scientific endeavor [1,2,3,4,5,6,8]. The findings from this study indicated that 2 weeks of mango ingestion did not mitigate plasma oxylipin levels after 2.25 h of intensive cycling.

## 2. Materials and Methods

### 2.1. Study Participants

Healthy male and female cyclists between the ages of 18 and 60 years who were capable of cycling at race pace for 2.25 h in a laboratory setting were recruited. Of the 45 adults assessed for eligibility, 24 were recruited into the study, and 22 (*n* = 13 males, *n* = 9 females) completed all study requirements. The two subjects that dropped out of the study experienced training-induced injuries. During the 6-week period when data were being collected, participants maintained their typical training regimen and did not ingest mangoes except for those provided during the study and avoided the use of supplements with vitamins and minerals, herbs, and medications that could influence inflammation. Participants signed informed consent forms, and study procedures were approved by the Institutional Review Board at Appalachian State University. Trial Registration: ClinicalTrials.gov, U.S. National Institutes of Health, identifier: NCT05409105. The research procedures were conducted at the Human Performance Laboratory operated by Appalachian State University at the North Carolina Research Campus (NCRC) in Kannapolis, NC.

Characteristics for the *n* = 22 study participants completing all aspects of the study protocol are summarized in Table 1. Male and female cyclists had similar ages, percentage body fats, and maximal oxygen consumption rates (VO_2max_). Male and female cyclists did not differ in exercise-induced changes in the primary outcomes for this study (total plasma oxylipins, supplement × time × sex interaction effect, and *p* = 0.520). Thus, data from this crossover study are presented for all participants combined.

### 2.2. Research Design

This study examined the effects of mango ingestion on moderating exercise-induced inflammation in a randomized crossover trial with 22 cyclists. In random order with trials separated by a 2-week washout period, the cyclists ingested 330 g of mango/day with 0.5 L water or 0.5 L of water alone for 2 weeks, followed by a 2.25 h cycling bout challenge. Blood and urine samples were collected pre- and post-2 weeks of supplementation, with additional blood samples collected immediately post-exercise and 1.5-h, 3-h, and 24 h post-exercise.

#### 2.2.1. Pre-Study Baseline Testing

After voluntarily signing IRB-approved consent forms, study participants were tested for maximal aerobic capacity (VO_2max_) during a graded cycling test with continuous metabolic monitoring with the Cosmed CPET metabolic cart (Cosmed, Rome, Italy). Body composition was measured with the BodPod body composition analyzer (Cosmed, Rome, Italy). Three-day food records and 24 h urine collection kits were supplied with thorough instructions. Participants were provided with a food list restricting high-fat foods with instructions to record all food and beverage intake (only from what was listed) during the 3-day period before the second lab visit. Macro- and micro-nutrient intake was assessed from the 3-day food records using the Food Processor dietary analysis software system (Version 11.11, ESHA Research, Salem, OR, USA).

#### 2.2.2. Pre-Supplementation Lab Visits

Study participants reported to the lab in an overnight fasted state and turned in their 3-day food records and 24 h urine samples. Blood samples were also collected. Study participants randomized to the mango trial were provided with a 2-week supply of frozen Tommy Atkins mangoes (pulp only in plastic freezer bags), with instructions to store the fruit in their freezers. Mangoes from Mexico were supplied, processed, and cubed by the National Mango Board (Orlando, FL, USA). The mango pulp was portioned into daily serving sizes and frozen. Participants were instructed to eat 165 g of mango/day with their first meal and then another 165 g with their last meal of the day during a 2-week period. Participants were allowed to eat the mango pulp as provided, mixed with yogurt, or in smoothies. Bottled water was supplied to all study participants with instructions to drink 0.5 L of water with the mangoes during the first meal and then another 0.5 L during the last meal. Participants were told to save the freezer bags and then bring them back to the next lab visit after the 2-week supplementation period. Participants randomized to the water-only trial were instructed to drink 0.5 L of water with the first meal and then another 0.5 L of water with the last meal. Participants crossed over to the other supplement after a 2-week washout interval during the second 2-week supplementation period. Participants were instructed to taper their exercise training during the 3-day period before the next lab visit to rest for the 2.25 h cycling session. Participants were also provided with another 3-day food record (with the food list), with instructions to record all food and beverage intake (only from what was listed) during the 3-day period before the next lab visit. A 24 h urine collection kit was also provided to participants with instructions to collect all urine during the entire day before the next lab visit.

#### 2.2.3. Cycling Sessions

Study participants reported to the lab in an overnight fasted state and turned in their 3-day food records, the empty mango freezer bags (to monitor compliance), and the 24 h urine sample. A blood sample was also collected. All participants drank 0.5 L of water, and those in the mango trial consumed 165 g of mango. The participants were not allowed to consume any other food or beverages.

After warming up, participants cycled for 2.25 h at an intensity close to a race of this duration (about 60% maximal watts power). Participants cycled on their own bicycles fitted to Saris H3 direct drive smart trainers (Madison, WI, USA) with monitoring by the Zwift online training platform (Long Beach, CA, USA). Heart rate, cycling speed, cadence, distance, power, breathing rate, and oxygen intake were measured after 15 min and then every 30 min during the cycling session using the cycling trainer, platform, and Cosmed metabolic cart. Participants ingested 250 mL water every 15 min with no other beverage or food allowed.

Performance data for the water-only and mango trials are summarized in Table 2. As designed, the two trials were similar in all performance measures, including cycling distance and speed, watts power output, oxygen consumption, and heart rate.

After completing the cycling session, additional blood samples were collected immediately post-exercise and then again after 1.5 and 3 h. During the first 1.5 h after the cycling session, participants drank water (7 mL/kg) and rested. After the blood draw and 1.5 h after exercise, participants in the mango trial consumed 165 g of mango. Participants in the water-only trial drank 0.45 L of a 6% carbohydrate sports beverage. Three hours after finishing the cycling session, participants were allowed to leave the performance lab. Participants were told to eat and drink beverages as desired from the assigned food list until the next morning and to avoid any additional vigorous exercise.

Participants returned the next morning in an overnight fasted state and provided a blood sample. For the next two weeks, participants were told to follow their normal eating habits and training routines. Participants then crossed over to the opposite trial arm and repeated all study procedures.

### 2.3. Sample Analysis

Plasma and urine aliquots were prepared and stored in a −80 °C freezer until analysis for oxylipins and mango phenolic metabolites after the study was completed.

#### 2.3.1. Plasma Oxylipins

Plasma arachidonic acid (ARA), eicosapentaenoic acid (EPA), docosahexaenoic acid (DHA), and oxylipins were analyzed using a liquid chromatography-multiple reaction monitoring mass spectrometry (LC-MRM-MS) method as fully described elsewhere [45]. Briefly, 100 µL of plasma sample were extracted with methanol-based protein precipitation at −20 °C overnight. The extracts werethen dried under nitrogen and reconstituted in 50 µL methanol/water (1:1) containing stable isotope labeled internal standards. Next, 10 µL of the reconstituted oxylipin samples were separated on a Waters HSS T3 column (100 × 2.1 mm, 1.8 μm) with a Vanguard pre-column (5 × 2.1 mm, 1.8 μm) and maintained at 40 °C and at a flow rate of 0.30 mL/min using a Thermo Vanquish UHPLC system and a binary mobile phase gradient (A: 0.1% formic acid in water; B: 0.1% formic acid in acetonitrile). The LC effluents were monitored using a Thermo Altis Plus triple quadrupole mass spectrometer with scheduled MRM mode for all the oxylipin analytes and internal standard transitions [45]. Resultant data files were processed with Skyline software (version 22.2.0.351), and the auto-integrated peaks were inspected manually. Concentrations of each oxylipin were determined from calibration curves of each analyte, which were constructed by normalizing to the selected deuterated internal standards followed by linear regression with 1/x weighting (Appendix A). Analytes with coefficients of variation relative to the quality control standards of <30% were included in the statistical analysis.

A summary composite variable was calculated for 49 of 67 oxylipins that had significant exercise-induced time effects. Based on prior studies from our research group [29,30], two other composite variables were calculated. Eight oxylipins generated from arachidonic acid and cytochrome P-450 (ARA-CYP) were grouped and these included 5,6-, 8,9-, 11,12-, and 14,15-dihydroxy-eicosatetraenoic acid (diHETrEs), 16-, 17,- 18-hydroxy-eicosatetraenoic acids (HETEs), and the 20-HETE metabolite 20-carboxy-arachidonic acid (20-coohAA). Four abundant oxylipins generated from linoleic acid with CYP and lipoxygenease (LOX) were also grouped, including 9,10- dihydroxy-9Z-octadecenoic acid (DiHOME), 12,13-DiHOME, 9- hydroxy-octadecadienoic acid (HODE), and 13-HODE (LA-DiHOMES + HODES).

#### 2.3.2. Urine Mango Metabolites

Creatinine concentrations in urine samples were quantified using an optical method that relies on the Jaffe reaction in a 96-well format (modified from reference [46]). Prior to mass spectrometry analysis, urine samples were centrifuged (15,000 rpm for 10 min), and the supernatants were diluted to a standard creatinine concentration of 2.5 mM.

UPLC–ESI–TOF analysis was performed using a Waters Xevo G2-XS QTOF mass spectrometer (Waters Corporation) coupled with an ACQUITY I-Class UPLC (Waters Corporation). A sample volume of 2 µL was separated on a Kinetex 2.6 µm PFP 100 Å 100 × 2.1 mm LC column (Phenomenex) maintained at 37 °C. A binary gradient using 0.1% formic acid in water (Mobile phase A) and 0.1% formic acid in acetonitrile (mobile phase B), from 2% B to 90% B over 15 min and flow rate gradient ranging from 0.55 mL/min to 0.75 mL/min, was utilized for separation (adapted from reference [47]).

The 12 mango metabolites analyzed were methylated, glucuronidated, or sulfated conjugates of pyrogallol, catechol, and gallic acid. These putative metabolites are not commercially available as chemical standards. Therefore, identification was based on accurate mass, fragmentation pattern, and literature precedence. Three structurally related standards with a relevant MS signal range (protocatechuic acid, gallic acid, and syringic acid) were utilized for quantitation based on three previous recent publications [48,49,50], which documented that these metabolites were significantly elevated following mango consumption. See Appendix A.

#### 2.3.3. Plasma Glucose

Plasma aliquots (pre-exercise, immediately post-exercise, and 1.5 h post-exercise) were analyzed for glucose concentrations using the protocol described in the glucose colorimetric assay kit (item no. 10009582) (Cayman Chemical, Ann Arbor, MI, USA).

### 2.4. Statistical Analysis

The data are expressed as mean ± SE. The plasma data were analyzed using the generalized linear model (GLM), repeated measures ANOVA module in SPSS (IBM SPSS Statistics, Version 28.0, IBM Corp, Armonk, NY, USA). The statistical model utilized the within-subjects approach: 2 (trials) × 6 (time points) repeated measures ANOVA and provided time (i.e., the collective effect of the cycling bouts) and interaction effects (i.e., whether the data pattern over time differed between trials). If the interaction effect was significant (*p* ≤ 0.05), post hoc analyses were conducted using paired *t*-tests comparing time point contrasts between trials. An alpha level of *p* ≤ 0.01 was used after Bonferroni correction for five multiple tests. The urine data were analyzed using the same approach but with a 2 (trials) × 2 (time points) repeated measure ANOVA. The positive false discovery rate (FDR or “*q* value”) was calculated for multiple testing corrections of the plasma oxylipin data.

## 3. Results

Three-day food records collected at the beginning and end of each 2-week supplementation period revealed no differences in macro- and micro-nutrient intake both within and between trials. The four 3-day food records were averaged, and for all 22 cyclists, the energy intake of the background diet averaged 1984 ± 106 kcal/day (8.3 ± 0.44 MJ/day), with carbohydrate, protein, fat, and alcohol representing 43.8 ± 1.3, 19.1 ± 0.9, 36.0 ± 0.8, and 2.0 ± 0.6%, respectively, of total energy.

Twelve putative gallotannin-derived metabolites were identified in the urine of subjects who consumed mango (Table 3). Three of these metabolites differed significantly between the mango and water-only (control) trials (Figure 1). These included *O*-methylgallic acid-*O*-sulfate (quantified against a syringic acid standard, interaction effect, *p* < 0.001), *O*-methylpyrogallol-*O*-sulfate (quantified against a syringic acid standard, *p* = 0.005), and pyrogallol-*O*-sulfate (quantified against a protocatechuic acid standard, *p* = 0.003).

Study participants who were randomized to mango ingestion consumed 165 g (~25 g carbohydrate) with water just before the 2.25 h cycling bout. Figure 2 shows that plasma glucose decreased following the cycling bouts for both supplements (mango and water-only) (*p* < 0.001) with no difference in the pattern of change (interaction effect, *p* = 0.543).

Plasma concentrations for ARA, EPA, and DHA increased significantly in response to the 2.25 h exercise bouts (all time effects, *p* < 0.001), but the pattern of change over time (interaction effects) was not significant (*p* values = 0.412, 0.380, and 0.740, respectively) (Figure 3).

A total of 67 oxylipins were detected in the study samples. Of these, 49 exhibited significant time effects during GLM statistical analysis (Appendix A). GLM 2 × 6 repeated measures analysis on each of these 49 oxylipins revealed no significant interaction effects. These 49 oxylipins were summed for a composite variable (Figure 4). GLM analysis showed a significant time effect (*p* < 0.001) of 2.25 h cycling on this composite variable of 49 oxylipins but without trial differences (interaction effect, *p* = 0.886). As explained in the methods section of this paper, two other composite variables were calculated that included eight pro-inflammatory oxylipins generated from arachidonic acid and cytochrome P-450 (ARA-CYP) and four abundant and pro-inflammatory oxylipins generated from linoleic acid (LA-DiHOMES + HODES) (Figure 4). Significant time effects were shown for ARA-CYP and LA-DiHOMES + HODES (*p* = 0.003 and *p* < 0.001, respectively), but the pattern of change over time did not differ between the mango and water-only trials (*p* = 0.610 and *p* = 0.168, respectively).

## 4. Discussion

In this randomized crossover clinical trial with 22 trained male and female cyclists, a 2-week intake of 330 g per day of mangoes increased urine concentrations of three targeted gallotannin-related gut-derived phenolics. Substantial increases in plasma concentrations of pro-inflammatory oxylipins were measured following 2.25 h of intensive cycling. However, contrary to our hypothesis, the pattern of change in these plasma oxylipins did not differ between the mango and water-only trials.

Gallotannins undergo catabolism by the gut microbiota, including hydrolysis and decarboxylation, to produce gallic acid, pyrogallol, and catechol. These colonic metabolite products are then absorbed and become subject to phase-II metabolism to produce methylated, glucuronidated, and sulfated conjugates. Previous studies investigating polyphenol metabolism after mango consumption found several of these metabolites were increased in the plasma or urine of subjects after acute, short-term (10 days) and long-term (6 weeks) consumption of mango [48,49,50]. Our study involved intake of a comparable mango dose for 2 weeks, and we expected to see similar results. Indeed, the three metabolites showing significant increases in our study, O-methylgallic acid-O-sulfate, O-methylpyrogallol-O-sulfate, and pyrogallol-O-sulfate (Figure 1), showed similar increases in the plasma and urine of subjects who consumed mango in the other trials [48,49,50].

Oxylipins are upstream regulators of many physiological processes, including inflammation. These oxidized lipids are typically produced from the actions of CYP, COX, and LOX enzyme systems on PUFAs such as LA, ARA, EPA, and DHA. Oxylipins are not stored but are generated during a variety of physiological stressors, including prolonged and vigorous exercise. Post-exercise plasma oxylipin concentrations are elevated for several hours, and the magnitude and duration of these responses shown in this study were similar to several other studies from our research group [5,7,29,30].

Oxylipin generation is a highly controlled process that is sensitive to both nutrition- and exercise-based interventions [51,52]. More than 50 enzymes are involved in producing oxylipins, and signaling is conducted by binding to a variety of receptors or by interacting with intracellular pathways [52,53,54]. The enzymes that regulate oxylipin production and the receptors to which they bind are potential targets for metabolites derived from dietary macronutrients and polyphenols [9,10,11,12,13,14,54]. Underlying mechanisms, however, are poorly understood, with large scientific gaps regarding the linkage between dietary change and oxylipin generation. In previous studies, our research group has discovered that the increase in ARA-CYP- and LA-DiHOMES + HODES-derived oxylipins were strongly mitigated when cyclists consumed carbohydrates (both 6% carbohydrate sports beverages and bananas) before, during, and after prolonged and intensive cycling bouts [5,29]. Carbohydrate supplementation may influence CYP activity within an exercise stress context by mitigating changes in plasma concentrations of glucose, insulin, and IL-6 [5,29]. In the current study, pre-cycling mango-carbohydrate intake (about 25 g) was insufficient to significantly alter plasma glucose levels (Figure 2).

More recently, in two studies, we reported that 2-weeks intake of 1 cup/day of blueberries mitigated plasma levels of ARA-CYP oxylipins in cyclists following 75 km cycling bouts and increased intermediate EPA- and DHA-derived specialized pro-resolving mediators (SPMs) during a 4-day recovery period from eccentric exercise in untrained adults [29,30]. Emerging evidence supports a regulatory effect of dietary polyphenols on the enzyme systems that are involved in oxylipin production, but many of these studies are based on cell culture methods with the parent molecules instead of the biotransformed metabolites [54]. Blueberries are rich in anthocyanins, and limited evidence indicates some linkage between gut-derived metabolites from these flavonoids and enzymatic activity related to selected oxylipin generation [29,30]. For example, we showed a modest but significant negative relationship between 1.5 h post-exercise plasma levels of blueberry metabolites (group of 24) and plasma levels of 10 ARA-CYP oxylipins [29]. One in vitro study showed a mild effect of pyrogallol (a metabolite from gallotannins as found in mangoes) on specific CYPs [55]. However, the actual blood and tissue levels of pyrogallol metabolites after mango ingestion may fall below the threshold needed to influence COX, LOX, and CYP-generated oxylipins, or the affected enzyme systems may not pertain to those activated in response to exercise stress. A potential limitation in the current study is the mango dosing regimen. The cyclists received 330 g/d mango for 2 weeks prior to the 2.25 h cycling sessions, with 165 g of mango ingested just prior to the cycling session. A longer supplementation period (e.g., 4 weeks), a higher pre-exercise mango dose, and the ingestion of 165 g of mango after approximately one hour of cycling may have a greater influence on post-exercise plasma oxylipin levels. Additionally, there are many other inflammation-based outcomes that were not measured in this study that could have been influenced by mango gut-derived metabolites (e.g., plasma cytokines and acute phase proteins).

## 5. Conclusions

This study showed that a large daily dose of mangoes over a 2-week supplementation period failed to alter exercise-induced increases in plasma oxylipins. These results differed from other studies using similar designs but with blueberries and bananas. Thus, mitigation of post-exercise plasma levels of oxylipins may depend on the specific macronutrient and polyphenol content of the fruit used during the 2-week pre-exercise intervention period. The practical implication for athletes is that not all fruits should be considered efficacious for their anti-inflammatory influences within an endurance exercise stress context.

## Figures and Tables

**Figure 1 nutrients-16-00036-f001:**
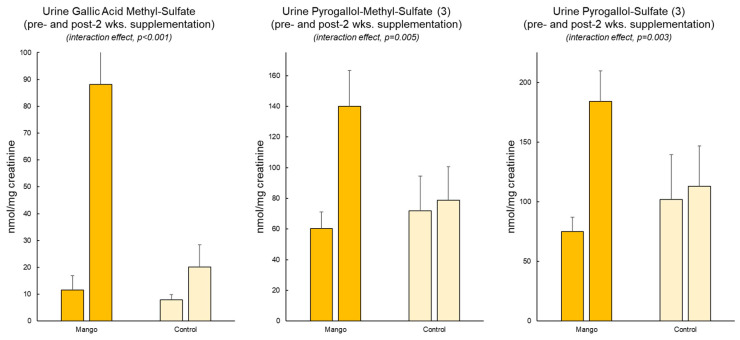
Increase in urine mango phenolic metabolites after 2 weeks of mango ingestion versus water-only (control). The data in this figure and all other figures are expressed as mean ± SE. SE was calculated as the standard deviation divided by the square root of the sample size.

**Figure 2 nutrients-16-00036-f002:**
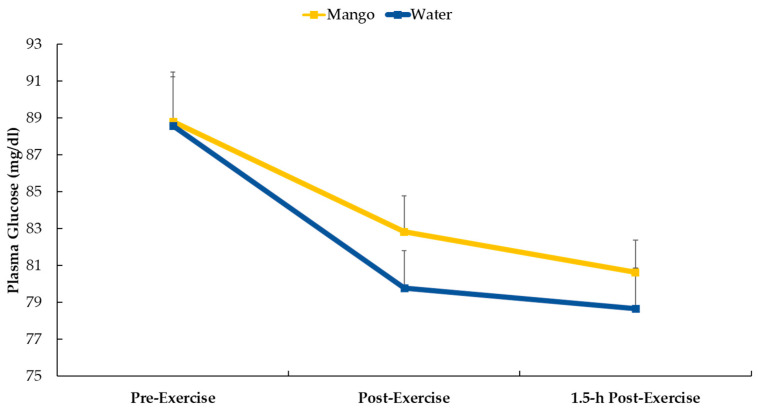
Plasma glucose data in the mango and water-only trials (interaction effect, *p* = 0.543).

**Figure 3 nutrients-16-00036-f003:**
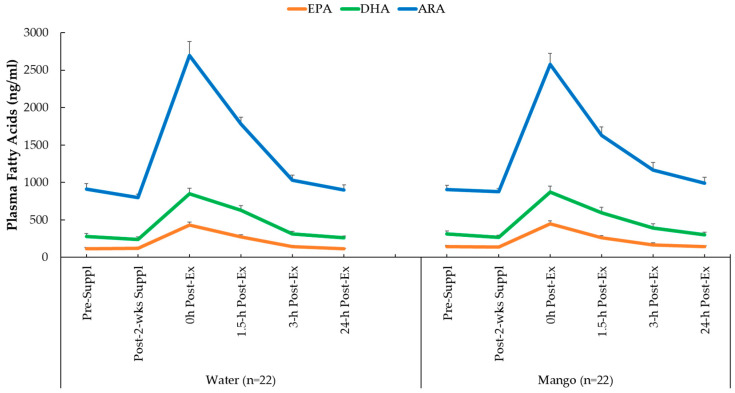
Changes in plasma concentrations for EPA, DHA, and ARA.

**Figure 4 nutrients-16-00036-f004:**
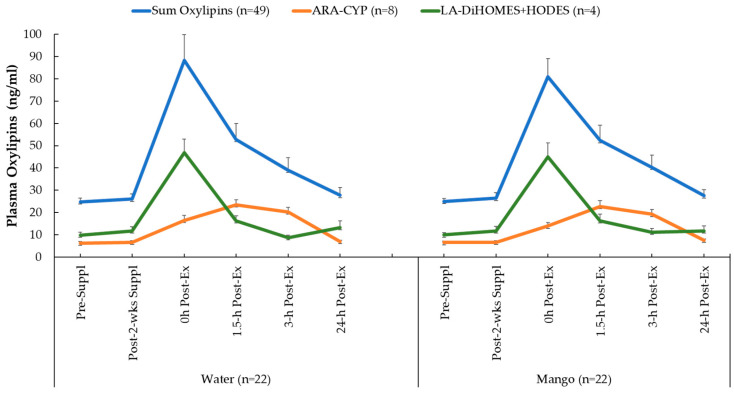
Changes in plasma oxylipin groups: sum of all 49 oxylipins with significant time effects; eight oxylipins generated from arachidonic acid and cytochrome P-450 (ARA-CYP); four abundant oxylipins generated from linoleic acid with CYP and LOX (9,10-DiHOME, 12,13-DiHOME, 9-HODE, 13-HODE) (LA-DiHOMES + HODES). See Appendix A for additional information.

**Table 1 nutrients-16-00036-t001:** Subject characteristics for male and female cyclists (*n* = 22).

Subject Characteristics Males (M), *n* = 13; Females (F), *n* = 9	Mean		Std. Error Mean	*t*-Test *p*-Value
Age (y)	M	43.15	2.12	0.169
	F	37.89	3.21	
Weight (kg)	M	84.08	2.27	<0.001
	F	59.03	2.09	
Height (cm)	M	181.31	1.37	<0.001
	F	166.06	1.72	
Body mass index (kg/m^2^)	M	25.56	0.55	<0.001
	F	21.44	0.84	
Body fat (%)	M	22.62	1.62	0.830
	F	23.19	2.15	
VO_2max_ (mL·kg^−1^min^−1^)	M	43.52	2.30	0.143
	F	37.94	2.87	
Watts max	M	282.69	13.69	<0.001
	F	197.22	12.80	
Heart rate max (beats/min)	M	169.85	2.29	0.966
	F	169.67	3.83	
Ventilation max (L/min)	M	124.78	9.60	0.005
	F	83.53	6.93	

**Table 2 nutrients-16-00036-t002:** Cycling performance measurements during the water-only and mango trials.

Performance Variable		Mean	*t*-Test *p*-Value
Distance cycled (km)	Water	60.9 ± 1.8	0.238
	Mango	59.1 ± 2.0	
Average speed (km/h)	Water	26.3 ± 0.8	0.325
	Mango	25.5 ± 0.9	
Average watts; % maximal watts	Water	145 ± 8.5 58.8 ± 1.9	0.756 0.907
	Mango	146 ± 9.0 59.0 ± 1.5	
Average VO_2_ (mL·kg^−1^min^−1^); % maximal VO_2_	Water	28.0 ± 1.2 68.6 ± 2.2	0.562 0.471
	Mango	28.5 ± 1.1 70.1 ± 2.3	
Average heart rate (beats/min); % maximal heart rate	Water	138 ± 3.4 71.0 ± 4.2	0.604 0.795
	Mango	136 ± 2.5 69.1 ± 3.8	

**Table 3 nutrients-16-00036-t003:** Putative gallotannin-derived metabolites characterized and quantified by Q-TOF in urine after consumption of mango. See Appendix A for additional information.

Putative Metabolite	Retention Time (min)	Calculated *m*/*z* (₋)	Measured *m*/*z* (₋)	MS/MS	Mass Error (ppm)
catechol-*O*-sulfate	2.14	188.9857	188.9854	109.0430	−1.59
*O*-methylcatechol-*O*-sulfate (isomer 1)	3.39	203.0014	203.0008	123.0531, 108.0317	−2.96
*O*-methylcatechol-*O*-sulfate (isomer 2)	4.59	203.0014	203.0006	123.0452	−3.94
*O*-methylcatechol-*O*-sulfate (isomer 3)	5.35	203.0014	203.0117	123.0522	50.74
*O*-methylgallic acid-*O*-glucuronide	1.85	359.0614	359.0617	168.0416, 312.9626	0.84
*O*-methylgallic acid-*O*-sulfate	2.99	262.9862	262.9866	183.0268, 168.0047	1.52
*O*-methylpyrogallol-*O*-sulfate (isomer 1)	1.56	218.9964	218.9956	139.043, 124.0129	−3.65
*O*-methylpyrogallol-*O*-sulfate (isomer 2)	2.07	218.9964	218.9957	139.0426, 204.9990, 125.0328	−3.20
*O*-methylpyrogallol-*O*-sulfate (isomer 3)	3.09	218.9964	218.9958	139.048, 124.0239	−2.74
pyrogallol-*O*-sulfate (isomer 1)	1.09	204.9807	204.9798	125.0255	−4.39
pyrogallol-*O*-sulfate (isomer 2)	2.05	204.9807	204.9799	125.0329	−3.90
pyrogallol-*O*-sulfate (isomer 3)	2.79	204.9807	204.9801	125.0329	−2.93

## Data Availability

The data presented in this study are available on request from the corresponding author. The primary outcome data (plasma oxylipins) are available as a Appendix A at https://www.mdpi.com/article/10.3390/nu16010036/s1.

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
