# Peer review of "Influence of 2 Weeks of Mango Ingestion on Inflammation Resolution after Vigorous Exercise"

_nutrients, 2023, doi:10.3390/nu16010036_

Round 1
Reviewer 1 Report
Comments and Suggestions for Authors
1. The experimental sample size is small, experimental errors may be caused by accidental factors, and the credibility is low.
2. The sample selection is only based on people aged 18-60 who can complete corresponding high-intensity cycling exercises, regardless of the age span of the sample. Different ages and genders have different abilities to digest and absorb mangoes, and the control of variables in the experiment was not strict enough.
3. Free selection of food in the diet during the experiment will affect the experimental results. The experimental group and the control group should eat the same type of food as the control group to eliminate the influence of irrelevant variables.
4. This experiment involves consuming 330 grams of mango per day. Is there a corresponding weight standard for choosing 330 grams of mango? There are three layers to choose from (low, medium and high), with different weights of mango intake.
5. After high-intensity cycling exercise, inflammation in the body may occur for several hours and peak between 24h and 72h. However, the inflammatory response in the experimental design was only detected in urine and blood 1.5h, 3h, and 24h after exercise, and the effect may not be obvious.
Comments on the Quality of English Language英语语言质量很好
Author Response
- The experimental sample size is small, experimental errors may be caused by accidental factors, and the credibility is low.RESPONSE: This is a randomized crossover trial, with n=22 subjects acting as their own controls. This design gives high statistical power, as we have shown in many published papers. See references 4,5, and PMID: 37025615, 32932235, 29566095, 29091464.
2. The sample selection is only based on people aged 18-60 who can complete corresponding high-intensity cycling exercises, regardless of the age span of the sample. Different ages and genders have different abilities to digest and absorb mangoes, and the control of variables in the experiment was not strict enough.
RESPONSE: The context for this study was to determine if mango ingestion could counter outcomes related to exercise stress. Thus, subjects had to be fit and capable of cycling for 2.25 h. We followed a randomized, crossover design. Subjects acted as their own controls. This is a high-level research design that reduces the influence of unmeasured, confounding factors.
3. Free selection of food in the diet during the experiment will affect the experimental results. The experimental group and the control group should eat the same type of food as the control group to eliminate the influence of irrelevant variables.
RESPONSE: Again, this was a randomized, crossover trial, with subjects acting as their own controls. Participants were given a food list restricting high fat foods with instructions to record all food and beverage intake (only from what was listed) during the 3-day period before the two cycling sessions. 3-day food records showed no differences between the mango and control trials for macro- and micro-nutrient intake.
4. This experiment involves consuming 330 grams of mango per day. Is there a corresponding weight standard for choosing 330 grams of mango? There are three layers to choose from (low, medium and high), with different weights of mango intake.
RESPONSE: The amount used in our study was similar to gram amounts in other human trials. For example, see PMID: 32109839.
5. After high-intensity cycling exercise, inflammation in the body may occur for several hours and peak between 24h and 72h. However, the inflammatory response in the experimental design was only detected in urine and blood 1.5h, 3h, and 24h after exercise, and the effect may not be obvious.
RESPONSE: We have shown in many exercise trials that the inflammatory response peaks immediately post-exercise to 1.5 h post-exercise, and is decreased significantly by 3h post-exercise (as shown in the oxylipin data in the current study). Inflammation is nearly resolved by 24 h. See Figure 5.
Reviewer 2 Report
Comments and Suggestions for Authors
In the article entitled "Influence of 2-Weeks Mango Ingestion on Inflammation Resolution After Vigorous Exercise”. The authors evaluated the anti-inflammatory effect of mango consumption on 22 cyclists for two weeks. The manuscript is well-written. However, there are some suggestions to further improve the manuscript:
1- The anti-inflammatory effect of mango is briefly mentioned in the introduction. Please extend it.
2- The resolution of almost all figures is very poor. The authors must improve it.
3- After Table 1 and Table 2. there is no space between the tables and the texts. Please introduce it.
4- The authors could emphasise the importance of their findings in affecting practice.
5- In the discussion section, the authors should mention the limitations of their study and how they can improve it in their future works.
6- In the conclusion section, it is advisable not to include references.
7- In the section on Author Contributions the phrase “For research articles with several authors, a short paragraph specifying their individual contributions must be provided. The following statements should be used” should be removed.
8- p. 1,3, line 16, 103: The term "in moderating" suggests that there must be "on moderating".
9- p. 1, 3, 4, lines 18, 106, 132: The term "water" suggests that there must be "of water".
10- p. 1, line 23: The term "mango" suggests that there must be "of mango".
11- p. 1, line 34: The term "are" suggests that there must be "is".
12- p. 2, line 66: The term "fruit" suggests that there must be "fruits".
13- p. 2, line 73: The term "plant" suggests that there must be "plants".
14- p. 2, line 83: The term "for quelling" suggests that there must be "to quelling".
15- p. 2, line 83: The term "intake" suggests that there must be a "the intake".
16- p. 2, line 90: The term "dropping" suggests that there must be a "dropped".
17- p. 2, line 92: The term "regimen, did" suggests that there must be a "regimen, and they did".
18- p. 2, line 83: The term "vitamin and mineral" suggests that there must be a "vitamins and minerals".
19- p. 4, line 148: The term "foods" suggests that there must be a "food".
20- p. 4, line 150: The term "cycled 2.25" suggests that there must be a "cycled for 2.25".
21- p. 4, line 167: The term "normal eating" suggests that there must be a "normal eating habit".
Comments on the Quality of English LanguageIn the manuscript entitled " Influence of 2-Weeks Mango Ingestion on Inflammation Resolution After Vigorous Exercise ". The authors have written the article in comprehensive English with minimal grammatical errors.
Author Response
In the article entitled "Influence of 2-Weeks Mango Ingestion on Inflammation Resolution After Vigorous Exercise”. The authors evaluated the anti-inflammatory effect of mango consumption on 22 cyclists for two weeks. The manuscript is well-written. However, there are some suggestions to further improve the manuscript:
1- The anti-inflammatory effect of mango is briefly mentioned in the introduction. Please extend it.
RESPONSE: Broadened this discussion in the introduction (see red text).
2- The resolution of almost all figures is very poor. The authors must improve it.
RESPONSE: Fixed this.
3- After Table 1 and Table 2. there is no space between the tables and the texts. Please introduce it.
RESPONSE: Added spaces.
4- The authors could emphasise the importance of their findings in affecting practice.
RESPONSE: Added more on practical implications in the discussion (see red text).
5- In the discussion section, the authors should mention the limitations of their study and how they can improve it in their future works.
RESPONSE: Added more on limitations in the discussion (see red text).
6- In the conclusion section, it is advisable not to include references.
RESPONSE: Removed these.
7- In the section on Author Contributions the phrase “For research articles with several authors, a short paragraph specifying their individual contributions must be provided. The following statements should be used” should be removed.
RESPONSE: Removed.
8- p. 1,3, line 16, 103: The term "in moderating" suggests that there must be "on moderating".
RESPONSE: Changed these as recommended.
9- p. 1, 3, 4, lines 18, 106, 132: The term "water" suggests that there must be "of water".
RESPONSE: Changed these as recommended.
10- p. 1, line 23: The term "mango" suggests that there must be "of mango".
RESPONSE: Changed this as recommended.
11- p. 1, line 34: The term "are" suggests that there must be "is".
RESPONSE: This sentence is based on "approaches" and this requires the use of "are".
12- p. 2, line 66: The term "fruit" suggests that there must be "fruits".
RESPONSE: Changed as follows: ..."by the intake of carbohydrates and polyphenols from fruits."
13- p. 2, line 73: The term "plant" suggests that there must be "plants".
RESPONSE: Changed to "plant foods".
14- p. 2, line 83: The term "for quelling" suggests that there must be "to quelling".
RESPONSE: Changed to "to counter".
15- p. 2, line 83: The term "intake" suggests that there must be a "the intake".
RESPONSE: Changed to "the intake".
16- p. 2, line 90: The term "dropping" suggests that there must be a "dropped".
RESPONSE: Changed to "that dropped".
17- p. 2, line 92: The term "regimen, did" suggests that there must be a "regimen, and they did".
RESPONSE: Added ..."they did..."
18- p. 2, line 83: The term "vitamin and mineral" suggests that there must be a "vitamins and minerals".
RESPONSE: Changed to "avoided the use of supplements with vitamins and minerals..."
19- p. 4, line 148: The term "foods" suggests that there must be a "food".
RESPONSE: Changed to..."The participants were not allowed to consume any other food or beverages."
20- p. 4, line 150: The term "cycled 2.25" suggests that there must be a "cycled for 2.25".
RESPONSE: Added "for."
21- p. 4, line 167: The term "normal eating" suggests that there must be a "normal eating habit".
RESPONSE: Changed to " normal eating habits and training routines."
Reviewer 3 Report
Comments and Suggestions for Authors
Dear authors of the study titled, "Influence of 2-Weeks Mango Ingestion on Inflammation Resolution After Vigorous Exercise "
Please note following points to finalize your manuscript for consideration in Nutrients Journal.
Corresponding author
Please assign
Abstract
OK
Key words
Add oxylipins, tandem mass spectrometry
Introduction
Overall the study background is presented well indicating relevant base work done by the authors. However, the rationale and the implications / novelty of the study is not presented. Please add a paragraph at the end of this section to clarify the above mentioned aspects.
L69-70. Mention the link in the references section.
Materials and Methods
The experiment design is well presented. However, the analytical results need improvement as per following points.
Figure 1 should be removed and written in the research design section.
2.3.1. Plasma Oxylipins
Briefly explain how the LC-MS/MS were reproduced based on Referene # 43.
2.3.2. Urine Mango Metabolites:
L198. TOF instruments can not perform MRM experiments. Replace it with UPLC–ESI–Q-TOF.
Results
L237. Table 1 should be moved to experimental section as a part of study design.
L247. Table 2 should be moved to experimental section.
L248. Start with new titled section/ paragraph.
L254. Table 3. It is unclear whether MRM or QTOF was used to obtain this data. Most probably it is later. Please add retention times, mass accuracy, quant data including concentrations, detection limits. Add technique used that resulted in this data.
Figure 2. These are your first results. In caption mention the statistical method created to get these error bars.
Discussion
OK
Conclusion
OK
Supplementary material
OK
Author contribution
Not evaluated
Acknowledgements
Not evaluated
References
OK
Author Response
Dear authors of the study titled, "Influence of 2-Weeks Mango Ingestion on Inflammation Resolution After Vigorous Exercise "
Please note following points to finalize your manuscript for consideration in Nutrients Journal.
1. Corresponding author, please assign.
RESPONSE: Now assigned.
2. Key words. Add oxylipins, tandem mass spectrometry
RESPONSE: Added.
3. Introduction: Overall the study background is presented well indicating relevant base work done by the authors. However, the rationale and the implications / novelty of the study is not presented. Please add a paragraph at the end of this section to clarify the above mentioned aspects.
RESPONSE: Added (see red text).
4. L69-70. Mention the link in the references section.
RESPONSE: Moved the links to the reference section.
5. Materials and Methods. The experiment design is well presented. However, the analytical results need improvement as per following points.
A. Figure 1 should be removed and written in the research design section.
RESPONSE: Removed. Description is in the text.
B. 2.3.1. Plasma Oxylipins. Briefly explain how the LC-MS/MS were reproduced based on Reference # 43.
RESPONSE: Brief description added regarding the LC-MS/MS analysis.
C. 2.3.2. Urine Mango Metabolites: L198. TOF instruments can not perform MRM experiments. Replace it with UPLC–ESI–Q-TOF.
RESPONSE: Changed this.
D. Results. L237. Table 1 should be moved to experimental section as a part of study design. L247. Table 2 should be moved to experimental section.
RESPONSE: Both tables moved to the methods section.
E. L248. Start with new titled section/ paragraph.
RESPONSE: Changed as described.
F. L254. Table 3. It is unclear whether MRM or QTOF was used to obtain this data. Most probably it is later. Please add retention times, mass accuracy, quant data including concentrations, detection limits. Add technique used that resulted in this data.
RESPONSE: Edited as recommended.
G. Figure 2. These are your first results. In caption mention the statistical method created to get these error bars.
RESPONSE: Added the method to calculate the SE bars.